# Metal and Ligand Effect on the Structural Diversity of Divalent Coordination Polymers with Mixed Ligands: Evaluation for Photodegradation

**DOI:** 10.3390/molecules28052226

**Published:** 2023-02-27

**Authors:** Manivannan Govindaraj, Shih-Ying Zhong, Chia-Her Lin, Jhy-Der Chen

**Affiliations:** 1Department of Chemistry, Chung-Yuan Christian University, Chung Li, Taoyuan City 320, Taiwan; 2Department of Chemistry, National Taiwan Normal University, Taipei 106, Taiwan

**Keywords:** coordination polymer, crystal structure analysis, dicarboxylate, coordination mode

## Abstract

Eight coordination polymers constructed from divalent metal salts, *N,N^′^*-bis(pyridin-3-ylmethyl)terephthalamide (**L**), and various dicarboxylic acids are reported, affording [Co(**L**)(5-ter-IPA)(H_2_O)_2_]_n_ (5-tert-H_2_IPA = 5-tert-butylisophthalic acid), **1**, {[Co(**L**)(5-NO_2_-IPA)]⋅2H_2_O}_n_ (5-NO_2_-H_2_IPA = 5-nitroisophthalic acid), **2**, {[Co(**L**)_0.5_(5-NH_2_-IPA)]⋅MeOH}_n_ (5-NH_2_-H_2_IPA = 5-aminoisophthalic acid), **3**, {[Co(**L**)(MBA)]⋅2H_2_O}_n_ (H_2_MBA = diphenylmethane-4,4′–dicarboxylic acid), **4**, {[Co(**L**)(SDA)]⋅H_2_O}_n_ (H_2_SDA = 4,4-sulfonyldibenzoic acid), **5**, {[Co_2_(**L**)_2_(1,4-NDC)_2_(H_2_O)_2_]⋅5H_2_O}_n_ (1,4-H_2_NDC = naphthalene-1,4-dicarboxylic acid), **6**, {[Cd(**L**)(1,4-NDC)(H_2_O)]⋅2H_2_O}_n_, **7**, and {[Zn_2_(**L**)_2_(1,4-NDC)_2_]⋅2H_2_O}_n_, **8**, which were structurally characterized by using single-crystal X-ray diffraction. The structural types of **1**–**8** are subject to the metal and ligand identities, showing a 2D layer with the **hcb**, a 3D framework with the **pcu**, a 2D layer with the **sql**, a polycatenation of 2-fold interpenetrated 2D layer with the **sql**, a 2-fold interpenetrated 2D layer with the 2,6L1, a 3D framework with the **cds**, a 2D layer with the 2,4L1, and a 2D layer with the (10^2^⋅12)(10)_2_(4⋅10⋅12^4^)(4) topologies, respectively. The investigation on the photodegradation of methylene blue (MB) by using complexes **1**–**3** reveals that the degradation efficiency may increase with increasing surface areas.

## 1. Introduction

Coordination polymers (CPs) have been intensively investigated by scientists in recent years because of their intriguing architectures and prospective applications in magnetism, luminescence, catalysis, gas storage, and sensing [1,2,3,4]. The coordination of spacer ligands to metal ions during the self-assembly process may result in the production of infinite one-dimensional (1D), two-dimensional (2D), or three-dimensional (3D) CPs, which are subject to the careful selection of metal ion and spacer ligands with diverse functionalities and flexibility. Despite the fact that many fascinating CPs have been reported, control of the structural variety remains a difficulty in the field of crystal engineering, and the factors that influence the structural diversity are less well understood [5,6].

The mixed ligand assembly technique has been employed to develop novel CPs [7]. In this context, mixed-ligand techniques including two distinct types of ligands with unique functions, such as polycarboxylate paired with a bis-pyridyl-bis-amide (bpba)-based N donor ligand, have been introduced as an effective way for adjusting structural diversity in CPs. Bpba ligands are remarkable ligands that may be modified to yield intriguing CPs [8], due to the fact that the majority of bpba ligands are flexible; however, others are semi-rigid.

Herein, we adopted the semi-rigid *N*,*N^′^*-bis(pyridin-3-ylmethyl)terephthalamide (**L**), as shown in Figure 1, and differently substituted dicarboxylic acid as part of our ongoing research into understanding the relationship between the mixed-ligand system and the structural variety of the new CPs. Several CPs containing **L** and dicarboxylate ligands have been structurally characterized. Three CPs, namely [Co(**L**)(HIPA)(H_2_O)_2_]⋅H_2_O (H_2_HIPA = 5-Hydroxyisophthalic acid), [Ni(**L**)(HIPA)(H_2_O)_2_]⋅H_2_O, and [Cu(**L**)(HIPA)]⋅4H_2_O, show similar 2D cellular networks for the first two complexes, and the latter has a 4-connected 3D structure with a semicircle 1D channel [9]. On the other hand, [Zn(**L**)(1,4-BDC)]·H_2_O (1,4-H_2_BDC = terephthalic acid), [Zn(**L**)(1,3- BDC)]·H_2_O (1,3-H_2_BDC = isophthalic acid), [Zn(**L**)(1,2-BDC)] (1,2-H_2_BDC = phthalate), and [Cd(**L**)_0.5_(1,2-BDC)(H_2_O)] display 2D structures with 6^3^ and 4^4^ topologies and a 4-connected 3D framework and a 1D structure, respectively [10], whereas [Cd_3_(**L**)_2_(1,4-bdc)_3_]·4H_2_O and [Cd(**L**)(1,4-bdc)]·2H_2_O are (3,5)-connected nets with the (3⋅7^2^ )(3^2^⋅4⋅7^5^⋅8^2^ ) topology [11], indicating that the types of the dicarboxylate play important role in determining the structural diversity.

The syntheses and crystal structures of [Co(**L**)(5-ter-IPA)(H_2_O)_2_]_n_ (5-tert-H_2_IPA = 5-tert-butylisophthalic acid), **1**, {[Co(**L**)(5-NO_2_-IPA)]⋅2H_2_O}_n_ (5-NO_2_-H_2_IPA = 5-nitroisophthalic acid), **2**, {[Co(**L**)_0.5_(5-NH_2_-IPA)]⋅MeOH}_n_ (5-NH_2_-H_2_IPA = 5-aminoisophthalic acid), **3**, {[Co(**L**)(MBA)]⋅2H_2_O}_n_ (H_2_MBA = diphenylmethane-4,4′–dicarboxylic acid), **4**, {[Co(**L**)(SDA)]⋅H_2_O}_n_ (H_2_SDA = 4,4-sulfonyldibenzoic acid), **5**, {[Co_2_(**L**)_2_(1,4-NDC)_2_(H_2_O)_2_]⋅5H_2_O}_n_ (1,4-H_2_NDC = naphthalene-1,4-dicarboxylic acid), **6**, {[Cd(**L**)(1,4-NDC)(H_2_O)]⋅2H_2_O}_n,_
**7**, and {[Zn_2_(**L**)_2_(1,4-NDC)_2_]⋅2H_2_O}_n_, **8**, form the subject of this report. We observed that the roles of the dicarboxylate ligands and the metal atoms in the structural diversity of the CPs prepared thusly are significant. The governing factors of **1**–**3** in the degradation of methylene blue (MB) were also evaluated.

## 2. Results and Discussion

### 2.1. Synthesis

Complexes **1**–**8** were prepared by the hydro(solvo)thermal reactions of **L** with corresponding dicarboxylic acids and metal salts in different solvent systems at 100 °C for 48 h. Hydro(solvo)thermal synthesis enables a unique combination of pressure and temperature for crystallization of CPs. Characteristic FT-IR peaks for complexes **1**–**8** are N-H and C=O stretching which are from L. The range of N-H stretching is 3386–3483 cm^−1^, probably coupled with the O-H stretching of the solvent molecule, while those around 1606–1653 cm^–1^ can be attributed to C=O stretching.

### 2.2. Crystal Structure of ***1***

The crystal structure of **1** conforms to the triclinic space group *P*ī and the asymmetric unit consists of one Co(II) cation, two halves of an **L** ligand, one 5-ter-IPA^2−^ ligand, and two coordinated water molecules. The Co(II) metal center is coordinated by two pyridyl nitrogen atoms of two **L** ligands [Co-N = 2.1346(12) − 2.1816(12) Å], two oxygen atoms from two 5-ter-IPA^2−^ ligands [Co-O = 2.0587(10) and 2.1362(10) Å], and two coordinated water molecules [Co-O = 2.0794(10) and 2.1411(10) Å], forming a distorted octahedral geometry, as in Figure 2a. Two Co(II) ions are bridged by two 5-ter-IPA^2−^ ligands to form dinuclear units, which are connected by **L** ligands to afford a 2D layer. Considering the Co(II) cations as 4-coordinated nodes, 5-ter-IPA^2^ as 2-connected nodes, and **L** ligands as linkers, the structure of **1** can be regarded as a 2,2,4-connected net with the point symbol (12)(4⋅12^5^)(4) (standard representation), as in Figure 2b, determined using ToposPro [12]. Moreover, if the dinuclear units are considered as 3-coordinated nodes, the structure can be further simplified as a 3-connected net with the (6^3^)-**hcb** topology (cluster representation) [13], as in Figure 2c.

### 2.3. Crystal Structure of ***2***

The structure of **2** was solved in the triclinic space group *P*ī with one Co(II) cation, two halves of an **L** ligand, one 5-NO_2_-IPA^2−^ ligand and two co-crystallized water molecules in each asymmetric unit. The Co(II) metal center is coordinated by four oxygen atoms from three 5-NO_2_-IPA^2−^ ligands [Co-O = 2.007(2) – 2.240(2) Å] and two pyridyl nitrogen atoms from two **L** ligands [Co-N = 2.149(3) – 2.150(3) Å], resulting in a distorted octahedral geometry, as in Figure 3a. Two Co(II) ions are bridged by two 5-NO_2_-IPA^2−^ ligands to form dinuclear units, which are connected by **L** ligands to afford a 3D framework. Considering the Co(II) cations as 5-coordinated nodes, 5-NO_2_-IPA^2−^ as 3-coordinated nodes, and **L** ligands as linkers, the structure of **2** can be regarded as a 3,5-connected binodal 3D net with the point symbol of (4^2^⋅6^5^⋅8^3^)(4^2^⋅6)-3,5T1 (standard representation), as in Figure 3b. Moreover, if the dinuclear units are considered as 6-coordinated nodes, the structure can be further simplified as a 6-connected net with the (4^12^⋅6^3^)-**pcu** topology (cluster representation), as in Figure 3c.

### 2.4. Crystal Structure of ***3***

Structural analysis demonstrates that **3** crystallizes in the triclinic *P*ī space group. The asymmetric unit contains one Co(II) cation, half of an **L** ligand, one 5-NH_2_-IPA^2−^, and one co-crystallized MeOH molecule. The Co(II) metal center is coordinated by four oxygen atoms from three 5-NH_2_-IPA ^2−^ ligands [Co-O = 2.0143(15)–2.2104(14) Å], one pyridyl nitrogen atom from the **L** ligand, and one nitrogen from the 5-NH_2_-IPA^2−^ ligand [Co-N = 2.1525(19)–2.2615(18) Å], showing a distorted octahedral geometry, as in Figure 4a. Two Co(II) ions are bridged by two 5-NH_2_-IPA^2−^ ligands to form dinuclear units, which are connected by **L** ligands to afford a 2D layer. Considering the Co(II) cations as 4-connected nodes and 5-NH_2_-IPA^2−^ ligands as 3-connected nodes, with **L** ligands as linkers, the structure of **3** can be simplified as a 3,4-connected 2D net with the {4^2^⋅6^3^⋅8}{4^2^⋅6}-**bey** topology (standard representation), as in Figure 4b. Moreover, if the dinuclear units are considered as 4-coordinated nodes, the structure can be further simplified as a 4-connected net with the (4^4^⋅6^2^)-**sql** topology (cluster representation), as in Figure 4c.

### 2.5. Crystal Structure of ***4***

Single crystal X-ray diffraction of **4** conforms to the orthorhombic space group *Ibca*, and the asymmetric unit consists of half of a Co(II) ion, half of an **L** ligand, half of an MBA^2−^ ligand, and one co-crystallized water molecule. Figure 5a shows the coordination environment around the Co(II) metal center, which is six coordinated by two nitrogen atoms from two **L** ligands [Co-N = 2.083(3)] and four oxygen atom from two MBA^2−^ ligands [Co-O = 2.064(2) – 2.263(3) Å], resulting in a distorted octahedral geometry. The Co(II) ions are interlinked by the **L** and MBA^2−^ ligands to give highly undulated 2D nets, as in Figure 5b. Topological analysis reveals that complex **4** forms 2-fold parallelly interpenetrated layers with the {4^4^⋅6^2^}-**sql** topology, as in Figure 5c. In addition, layers of the 2-fold interpenetrated 2D layers polycatenated with other **sql** layers to form a final 2D → 3D entanglement, as in Figure 5d.

### 2.6. Crystal Structures of ***5***

In the space group *P*ī, the structure of complex **5** was solved. The asymmetric unit consists of one Co(II) ion, half of an **L** ligand, one SDA^2−^ ligand, and one co-crystallized water molecule. Figure 6a shows the coordination environment around the dinuclear Co(II) centers with a Co---Co distance of 2.8143(5). Both Co(1) and Co(2) are 5-coordinated by one pyridyl nitrogen atom of the **L** ligand [Co–N = 2.0564(17) Å] and four oxygen atoms of four SDA^2−^ ligands [Co–O = 2.0211(17) Å–2.0515(17) Å], resulting in a distorted square pyramidal geometry. Two Co(II) ions are bridged by four carboxylate groups of the SDA^2−^ ligands to form dinuclear paddlewheel units, which are further linked by the **L** ligands to form a 2D layer. If the dinuclear units are considered as 6-connected nodes, the SDA^2−^ ligands as 2-connected nodes, and the **L** ligands as linkers, the structure of **5** can be simplified as a 2D net with the (4^2^⋅6^8^⋅8⋅10^4^)(4)_2_-2,6L1 topology, as in Figure 6b, which shows a 2-fold interpenetration, as in Figure 6c.

### 2.7. Crystal Structures of ***6***

Crystals of **6** conform to the monoclinic space group *P*2_1_/*c* with each asymmetric unit consisting of one and two halves of a Co(II) cation, two **L** ligands, two 1,4-NDC^2−^ ligands, two coordinated water molecules, and five co-crystallized water molecules. Figure 7a shows the coordination environment of the Co(II) metal centers, which are all 6-coordinated. The Co(1) atom is coordinated by two nitrogen atoms from two **L** ligands [Co-N = 2.134(2) and 2.138(2) Å], three oxygen atoms from two 1,4-NDC^2−^ ligands [Co–O = 2.0257(18) Å – 2.19740515(19) Å], and one oxygen atom from the coordinated water molecule [Co-O = 2.0817(19) Å]. The Co(2) atom is located at the inversion center, which is coordinated by two pyridyl nitrogen atoms [Co–N = 2.114 (2) Å] from two **L** ligands and four oxygen atoms from two 1,4-NDC^2−^ ligands [Co–O = 2.1128(17)–2.1415(18) Å], whereas the Co(3) atom, which is also located at the inversion center, is coordinated by two nitrogen from two **L** ligands [Co–N = 2.145(2) Å], two oxygen atoms from two 1,4-NDC^2−^ ligands [Co–O = 2.0813(17)] and two oxygen atoms from two water molecules [Co–O = 2.1497(18)]. The Co(II) ions are linked by the 1,4-NDC^2−^ and **L** ligands to form a 3D framework. If the Co(II) ions are defined as 4-connected nodes and the **L** and 1,4-NDC^2−^ ligands as linkers, the structure of **6** can be simplified as a 4-connected net with the (6^5^⋅8)-**cds** topology, as in Figure 7b.

### 2.8. Crystal Structures of ***7***

Single-crystal X-ray diffraction analysis shows that **7** crystallizes in the monoclinic space group *P*2_1_/*c*. The asymmetric unit is comprised of one Cd(II) cation, one **L** ligand, one 1,4-NDC^2−^ ligand, one coordinated water, and two lattice water molecules. Figure 8a depicts a drawing showing the coordination environment of the Cd(II) ion, which is 7-coordinated by four oxygen atoms from two 1,4-NDC^2−^ ligands, one oxygen atom from the water molecule [Cd–O = 2.343(2) − 2.396(19) Å], and two pyridyl nitrogen atoms from two **L** ligands [Cd–N = 2.319(2) and 2.410(3) Å]. The Cd (II) ions are further linked together by the **L** and 1,4-NDC^2−^ ligands to afford a 2D layer. If the Cd(II) cations are defined as 4-connected nodes and the 1,4-NDC^2−^ and **L** ligands are defined as 2-connected nodes, the structure of **7** can be simplified as a 2D net with the {4⋅5^8^}{4}-2,4L1 topology, as in Figure 8b.

### 2.9. Crystal Structures of ***8***

The crystals of complex **8** conform to the triclinic space group *P*ī with two Zn(II) ions, two **L** ligands, two 1,4-NDC^2−^ ligands, and two lattice water molecules in the asymmetric unit. Figure 9a shows the coordination environment of the Zn(II) centers. Both of the Zn(1) and Zn(2) atoms form tetrahedral geometries, which are 4-coordinated by two nitrogen atoms [Zn(1)–N(1) = 2.066(3); Zn(1)–N(4C) = 2.062(3) Å; Zn(1)–N(5) = 2.056(3) Å; Zn(1)–N(8A) = 2.066(3) Å] from two **L** ligands and two oxygen atoms [Zn(1)–O(5) = 1.967(3) Å; Zn(1)–O(9) = 1.975(2); Zn(1)–O(7) = 1.969(2) Å; Zn(1)–O(12B) = 1.948(2)] from two 1,4-NDC^2−^ ligands. Topological analysis demonstrates that complex **8** displays a 2D net with the point symbol of (10^2^⋅12)(10)_2_(4⋅10⋅12^4^)(4), as in Figure 9b.

### 2.10. Ligand Conformations and Coordination Modes

The **L** ligands in complexes **1**–**8** display various conformations which can be defined as follows: (A) the *cis* and *trans* conformations can be given if the two C=O groups are in the same and the opposite direction, respectively; (B) due to the different orientations adopted by the pyridyl nitrogen atoms and the amide oxygen atoms, three more conformations, namely *syn–syn*, *syn–anti*, and *anti*–*anti*, can also be found for bpba [8]. Table 1 lists the ligand conformations and coordination modes of the organic ligands in complexes **1**–**8**. The **L** ligands in **1**–**8** bridge two metal ions through two pyridyl nitrogen atoms, adopting five different conformations including *trans anti–anti*, *cis syn–syn*, *trans syn–syn*, *trans syn–anti* and *cis anti–anti*. On the other hand, the dicarboxylate ligands in **1**–**8** bridge two to four metal ions with various coordination modes.

### 2.11. Structural Comparisons

Structural comparisons of complexes **1**–**8** show that the structural diversity is subject to the change in the dicarboxylate ligand. The different structural types in **1**–**3** demonstrate the substituent effect of the group at the fifth position of the phenyl ring. The use of the angular dicarboxylic acids, such as H_2_MBA and H_2_SDA, give entangled **4** and **5**, showing a polycatenated net of 2-fold interpenetration and a 2-fold interpenetrated net, respectively. The metal effect on the structural diversity is shown in **6**–**8** by changing the metal atom from Co, Cd, to Zn, giving **cds**, 2,4L1, and (10^2^⋅12)(10)_2_(4⋅10⋅12^4^)(4) topologies, respectively.

### 2.12. Photodegradation

The governing role of CPs in the photodegradation of organic pollutants has been a subject of current interest [14,15,16,17,18,19]. Complexes **1**–**3**, which differ in the fifth position of the phenyl ring of the dicarboxylate ligands, *i.e.*, the *tert*-butyl, NO_2_ and NH_2_ groups, respectively, thus, provide a unique opportunity to compare the substituent effect on the photodegradation. Methylene blue (MB, C_16_H_18_ClN_3_S) was selected as the dye contaminant, and the experiments were carried out with 30 wt % H_2_O_2_ under 365 nm UV light. Time-dependent absorption spectra of the MB solutions under 365 nm UV light are provided as Appendix A.

The intensity of the peculiar absorption band at 663 nm was utilized to precisely monitor the degradation process of MB. Figure 10 illustrates the variations in the A_t_/A_0_ of MB solutions vs irradiation time for complexes **1–3**, showing that the absorption intensities of MB reduced gradually with increasing reaction time, where A_0_ is the initial absorbance of the MB solution and A_t_ is the absorbance of the solution after illumination at time t. Degradation efficiency (DE) of MB was calculated by using DE % = [(A_0_ − A_t_)/A_0_] × 100. Additionally, the DE % with the mean values and standard deviations were evaluated, Appendix A. After 120 min, the DE of MB for the various strategies are as follows: 3% (blank), 53.46% (MB + H_2_O_2_), 6% (MB + complex **1**), 10.42% (MB + complex **2**), 16.43% (MB + complex **3**), 61.35% (MB + H_2_O_2_ + complex **1**), 77.59% (MB + H_2_O_2_ + complex **2**), and 95.06% (MB + H_2_O_2_ + complex **3**), demonstrating that the DE % of MB by the complexes participated with H_2_O_2_ follows the pattern of **1** < **2** < **3**. Moreover, the Brunauer–Emmett–Teller (BET) surface areas obtained from the N_2_ adsorption experiments were 4.96, 6.12, and 9.95 m²/g for **1**–**3**, respectively, as in Appendix A. The PXRD patterns of complexes **1**–**3** succeeding photodegradation processes were examined. No noticeable alterations were found for **1** and **2**, whereas significant change has been observed for **3**, as illustrated in Appendix A. The structural change in **3** may enhance the photodegradation efficiency. Structural modification was also observed for **3** after the N_2_ adsorption and desorption, as in Appendix A, indicating that complex **3** was not stable during the experiments.

Although the role of the *tert*-butyl, NO_2_, and NH_2_ groups in determining the DE is complicated, the different BET surface areas of the original **1**–**3** resulting from the different substituent groups can be influential. The hydroxyl radical (OH⋅) has been considered as the major oxidant which decomposes the organic dye with a good efficiency [15]. High surface area reflects a higher adsorption quantity of H_2_O_2_ that led to the formation of (OH⋅) and, thus, implies more MB can be degraded. For comparisons it is noted that the CPs {[Zn(L2)(AIPA)]·2H_2_O}_n_ (L2 = *N,N′*-bis(3-pyridinyl)terephthalamide; H_2_AIPA = 5-acetamidoisophthalic acid) and {[Zn(L3)(AIPA)]·2H_2_O}_n_ (L3 = *N,N′*-di(3-pyridyl)adipoamide), which adopted self-catenated 3D frameworks with the (4^24^·6^4^)-8T2 and the (4^4^·6^10^·8)-**mab** topologies, respectively, promoted the MB degradation, and the DE were 81.56 and 85.46%, respectively [20]. On the other hand, the four topologically identical CPs having the 2-fold interpenetrating 3D net with the **mog** topology, {[M(L4)_0.5_(L5)(H_2_O)_2_]⋅H_2_O}_n_ (M = Co and Ni; H_4_L4 = bis(3,5-dicarboxyphenyl)adipoamide; L5 = bis(N-pyrid-3-ylmethyl) adipoamide) and {[M_2_(L4)(L6)_2_(H_2_O)_4_]⋅3H_2_O}_n_ (M = Co and Ni; L6 = bis(N-pyrid-3-ylmethyl) suberoamide) also display good photodegradation performance toward MB, and the Co(II) CPs display better catalytic ability than the Ni(II) ones [21].

## 3. Materials and Methods

### 3.1. General Procedures

Elemental analyses involving C, H, and N atoms were performed on a PE 2400 series II CHNS/O (PerkinElmer instruments, Shelton, CT, USA) or an Elementar Vario EL-III analyzer (Elementar Analysensysteme GmbH, Hanau, Germany). Infrared spectra were obtained from a JASCO FT/IR-460 plus spectrometer with pressed KBr pellets (JASCO, Easton, MD, USA). Powder X-ray diffraction patterns were carried out with a Bruker D8-Focus Bragg–Brentano X-ray powder diffractometer equipped with a CuKα (λ_α_ = 1.54178 Å) sealed tube (Bruker Corporation, Karlsruhe, Germany).

### 3.2. Materials

The reagents Co(OAc)_2_·4H_2_O, Cd(OAc)_2_·H_2_O and Zn(OAc)_2_·2H_2_O were purchased from Alfa Aesar (Ward Hill, MA, USA), whereas 5-tert-butylisophthalic acid (5-tert-H_2_IPA), 5-nitroisophthalic acid (5-NO_2_-H_2_IPA), 5-aminoisophthalic acid (5-NH_2_-H_2_IPA ), diphenylmethane-4,4′–dicarboxylic acid (H_2_MBA ), 4,4-sulfonyldibenzoic acid (H_2_SDA), and naphthalene-1,4-dicarboxylic acid (1,4-H_2_NDC) were from Aldrich Chemical Co. (St. Louis, MO, USA). The ligand *N,N′*-bis(pyridin-3-ylmethyl)terephthalamide (**L**) was prepared according to a published procedure [22].

### 3.3. Preparations

#### 3.3.1. [Co(**L**)(5-ter-IPA)(H_2_O)_2_]_n_, **1**

A 23 mL Teflon-lined steel autoclave was sealed with Co(OAc)_2_⋅4H_2_O (0.050 g, 0.20 mmol), **L** (0.070 g, 0.20 mmol), 5-tert-H_2_IPA (0.042 g, 0.20 mmol), and 10mL H_2_O, which was heated to 100 °C for two days and then cooled to room T at a rate of 2 °C per hour. Orange crystals formed, which were collected and purified. Yield: 0.061 g (46%). Anal. calcd for C_32_H_34_CoN_4_O_8_ (MW = 661.56): C, 58.1; H, 5.1; N, 8.5%. Found: C, 58.8; H, 5.3; N, 8.7%. FT-IR (cm^−1^): 3420(s), 2965(w), 1663(m), 1606(m), 1536(m), 1479(m), 1431(m), 1372(m), 1281(w), 1187(w), 1107(m), 1038(w), 933(m), 786(s), and 710(s).

#### 3.3.2. {[Co(**L**)(5-NO_2_-IPA)]⋅2H_2_O}_n,_
**2**

Purple crystals of **2** were prepared by following similar procedures for **1**, except that 5-NO_2_-H_2_IPA (0.042 g, 0.20 mmol) was used. Yield: 0.065 g (50%). Anal. calcd for C_28_H_25_CoN_5_O_10_ (MW = 650.46): C, 51.7; H, 3.9; N, 10.8%. Found: C, 52.2; H, 3.6; N, 10.8%. FT-IR (cm^−1^) :3473(s),3386(s),3263(s),3083(m), 1647(m), 1613(m), 1556(m), 1530(m), 1464(m), 1392(m), 1346(m), 1288 (w), 1083(w), 1038(w), 996(m), 736(s), 718(s), and 702(s).

#### 3.3.3. {[Co(**L**)_0.5_(5-NH_2_-IPA)]⋅MeOH}_n_, **3**

Complex **3** was prepared by following similar procedures for **1**, except that 5-NH_2_-H_2_IPA (0.036 g, 0.20 mmol) in 10 mL of MeOH/H_2_O was used. Red crystals were collected. Yield: 0.064 g (72%). Anal. calcd for C_19_H_18_CoN_3_O_6_ (MW = 443.29): C, 51.5; H, 4.1; N, 9.5%. Found: C, 51.2; H, 3.7; N, 9.5%. FT-IR (cm^−1^): 3389(s),3326(s), 1653(m), 1546(m), 1530(m), 1451(m), 1404(m), 1346(m), 1278 (w), 1055(w), 1033(w), 957(m), 781(s), 729 (s), and 711(s).

#### 3.3.4. {[Co(**L**)(MBA)]⋅2H_2_O} _n_, **4**

Complex **4** was prepared by following similar procedures for **1**, except that a mixture of Co(OAc)_2_⋅4H_2_O (0.05 g, 0.20 mmol), **L** (0.070 g, 0.20 mmol), and MBA (0.052 g, 0.20 mmol) in 10 mL of H_2_O was used. Purple crystals were obtained. Yield: 0.084 g (60%). Anal. calcd for C_35_H_32_CoN_4_O_8_ (MW = 695.57): C, 60.4; H, 4.6; N, 8.1%. Found: C, 60.7; H, 4.3; N, 7.7%. FT-IR (cm^−1^): 3450 (s), 2920(m),2850(m), 1629(m), 1613(m), 1469(s), 1392(m), 1358(m), 1301(w), 871(m), 760(s), 727(w), and 702(w).

#### 3.3.5. {[Co(**L**)(SDA)]⋅H_2_O}_n ,_
**5**

Complex **5** was prepared by following similar procedures for **1**, except that a mixture of Co(OAc)_2_.4H_2_O (0.025 g, 0.10 mmol), **L** (0.035 g, 0.10 mmol), SDA (0.031 g, 0.10 mmol), and 8 mL of H_2_O in 2 mL of MeOH was used. Violet crystals were obtained. Yield: 0.023 g (41%). Anal. calcd for C_24_H_19_CoN_2_O_8_S (MW = 554.40): C, 52.0; H, 3.4; N, 5.1%. Found: C, 51.9; H, 3.4; N, 6.5%. The large inconsistency of the N atom may be due to the fact that the crystals used for measurement suffered the loss of co-crystallized solvents or the contamination of minor product which was not able to be removed. FT-IR (cm^−1^): 3428(s), 2925(s), 2851(m), 3083(m), 1648(m), 1549(m), 1418(m), 1301(m), 1173(m), 1127(m), 1035(w), 1288 (w), 992(w), 862(w), 762 (s), and 700(s).

#### 3.3.6. {[Co_2_(**L**)_2_(1,4-NDC)_2_(H_2_O)_2_]⋅5H_2_O}_n_, **6**

Complex **6** was prepared by following similar procedures for **1**, except that a mixture of Co(OAc)_2_·4H_2_O (0.025 g, 0.10 mmol), 1,4-H_2_NDC (0.022 g, 0.10 mmol), and **L** (0.035 g, 0.10 mmol) in 10 mL H_2_O was used, and the reaction was carried out at 80 °C. Pink crystals were obtained. Yield: 0.055 g (81%). Anal. calcd for C_64_H_62_Co_2_N_8_O_19_ (MW = 1365.07): C, 56.3; H, 4.6; N, 8.2%. Found: C, 56.8; H, 4.3; N, 8.8%. IR (cm^−1^): 3483(s), 3314(s), 2915(m), 1649(s), 1546(s), 1428(m), 1349(m), 1295(m), 1256(w), 1198(w), 1051(m), 977(s), and 838(m).

#### 3.3.7. {[Cd(**L**)(1,4-NDC)(H_2_O)]⋅2H_2_O}_n_, **7**

Prepared as described for **6**, except that Cd(OAc)_2_·H_2_O (0.027 g, 0.10 mmol) was used. Colorless crystals were obtained. Yield: 0.039 g (54%). Anal. calcd for C_32_H_30_CdN_4_O_9_ (MW = 727.00): C, 52.7; H, 4.1; N, 7.7%. Found: C, 52.7; H, 3.8; N, 7.5%. IR (cm^−1^): 3436(s), 1643(s), 1559(m), 1428(s), 1367(s), 1284(m), 1234(m), 1189(m), 1117(w), 1058(w), 873(m), 846(s), and 700(m).

#### 3.3.8. {[Zn_2_(**L**)_2_(1,4-NDC)_2_]⋅2H_2_O}_n_, **8**

Complex **8** was prepared by following similar procedures to those for **6**, except that a mixture of Zn(OAc)_2_·2H_2_O (0.022g, 0.10 mmol) was used. Colorless crystals were obtained. Yield: 0.061 g (95%). Anal. calcd for C_64_H_52_Zn_2_N_8_O_14_ (MW = 1287.87): C, 59.7; H, 4.1; N, 8.7%. Found: C, 60.4; H, 3.7; N, 8.7%. IR (cm^−1^): 3330(s), 3060(m), 2929(w), 1644(s), 1600(s), 1540(s), 1430(m), 1328(m), 1291(w), 1257(w), 1193(m), 1123(w), 833(m), 786(m), and 701(m).

### 3.4. Powder X-ray Analysis and IR Spectra

In order to check the phase purity of the product, powder X-ray diffraction (PXRD) experiments were carried out for complexes **1**–**8**. As shown in Appendix A, the peak positions of the experimental and simulated PXRD patterns were in a good agreement with each other, indicating their bulk purities. The IR spectra of complexes **1**–**8** are provided in the Appendix A as Appendix A.

### 3.5. Procedures for Photodegradation

The experiments were carried out in a homemade photodegradation box (Appendix A). For the experiments, test tube 1 (blank), tube 2 (0.1 mL H_2_O_2_), tube 3 (10 mg complex), and tube 4 (10 mg complex + 0.1 mL H_2_O_2_) were prepared. A total of 10 mL of a 10 ppm MB solution was added to each tube, which was prepared by diluting 10 mg MB with deionized water in a 1000 mL quantitative bottle. Each tube was then irradiated with the 365 nm UV light for 20, 40, 60, 80, 100, and 120 min, respectively, and then their absorption spectra were measured. Tube 3 and tube 4 were first stirred in the dark for 15 min to confirm the physical adsorption of the complex.

### 3.6. X-ray Crystallography

Single-crystal X-ray diffraction data for complexes **1**–**8** were collected on a Bruker AXS SMART APEX II CCD diffractometer with graphite-monochromated MoKα (λ_α_ = 0.71073 Å) radiation at 296 K [23]. Data reduction and absorption correction were performed by using standard methods with well-established computational procedures. Some of the heavier atoms were located by the direct or Patterson method, and the remaining atoms were found in a series of Fourier maps and least-squares refinements, while the hydrogen atoms were added by using the HADD command in SHELXTL [24]. Table 2 lists the basic information pertaining to crystal parameters and structure refinement. CCDC no. 2238099–2238106 contain the supplementary crystallographic data for this paper. These data can be obtained free of charge via http://www.ccdc.cam.ac.uk/conts/retrieving.html or from the Cambridge Crystallographic Data Centre, 12 Union Road, Cambridge CB2 1EZ, UK; fax: +44 1223 336 033; e-mail: deposit@ccdc.cam.ac.uk; or at: http://www.ccdc.cam.ac.uk.

## 4. Conclusions

Eight divalent CPs constructed from **L** and various dicarboxylic acids have been successfully accomplished. The changes in the substituted group at the fifth position of the phenyl rings of the dicarboxylic acids from *tert*-butyl and NO_2_ to the NH_2_ group drastically alters the structural types, affording the simplified structures with the **hcb**, **pcu**, and **sql** topologies for complexes **1**–**3**, respectively. The use of the angular dicarboxylic acids, such as H_2_MBA and H_2_SDA, gave entangled CPs **4** and **5**, showing a polycatenation of a 2-fold interpenetrated 2D layer with the **sql** and a 2-fold interpenetrated 2D layer with the 2,6L1 topologies, whereas the metal effect on the structural diversity can be shown in complexes **6**–**8** by changing the metal atom from Co, Cd to Zn, affording a 3D framework with the **cds**, a 2D layer with the 2,4L1, and a 2D layer with the (10^2^⋅12)(10)_2_(4⋅10⋅12^4^)(4) topologies, respectively. The structural diversity of the semi-rigid **L**-based divalent CPs is, thus, subject to the identities of the metal atom and the dicarboxylic acid. The degradation efficiency toward MB that follows **1** < **2** < **3** can be ascribed to their increasing surface areas, resulting from the different substituent groups of *tert*-butyl, NO_2_, and NH_2_ at the fifth position of the phenyl ring of the respective dicarboxylate ligands.

## Figures and Tables

**Figure 1 molecules-28-02226-f001:**
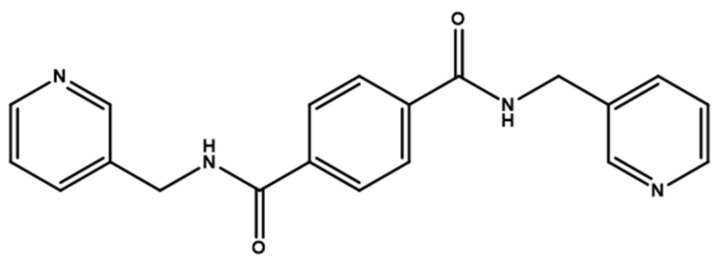
Structure of **L**.

**Figure 2 molecules-28-02226-f002:**
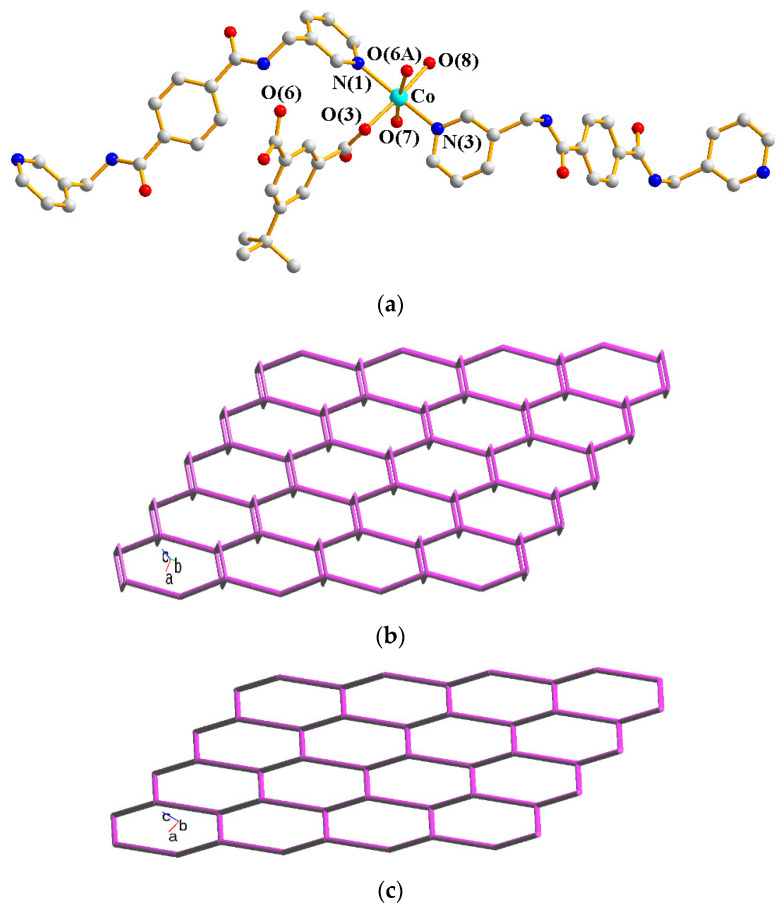
(**a**) A drawing showing the geometry of the Co(II) ion in **1**. Symmetry transformations: (A) –x + 1, −y + 1, −z. (**b**) A drawing showing the (12)(4⋅12^5^)(4) topology. (**c**) A drawing showing the **hcb** topology.

**Figure 3 molecules-28-02226-f003:**
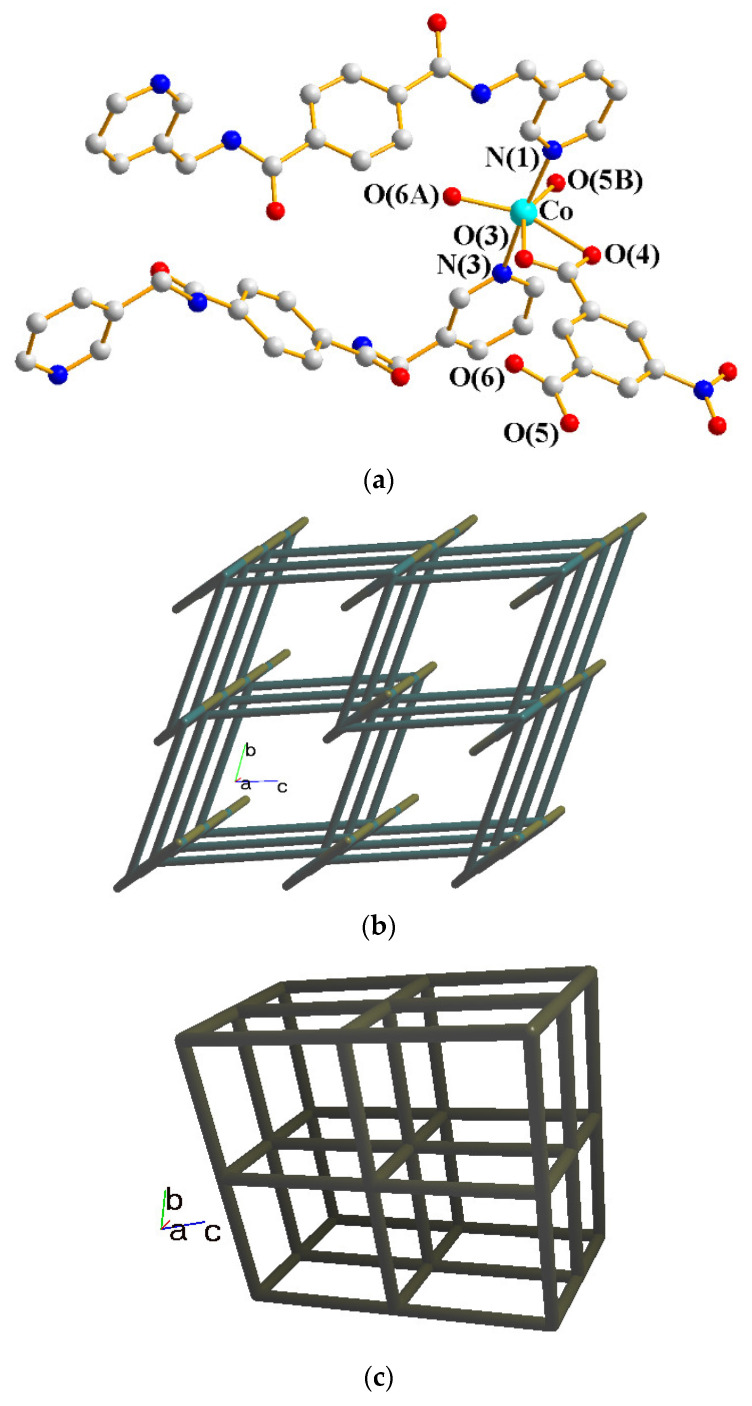
(**a**) Coordination environment of the Co(II) ion in **2**. Symmetry transformations: (A) −x + 1, −y, −z; (B) x + 1, y, z. (**b**) A drawing showing the 3,5T1 topology. (**c**) A drawing showing the **pcu** topology.

**Figure 4 molecules-28-02226-f004:**
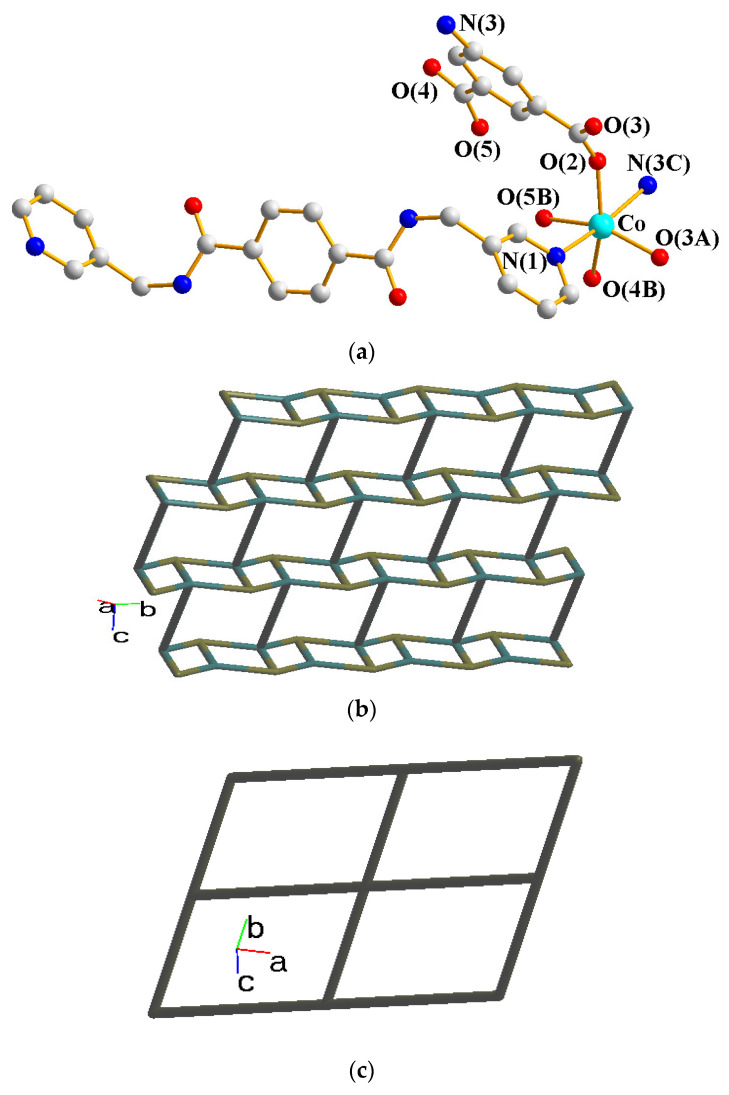
(**a**) Coordination environment about the Co(II) ion in **3**. Symmetry transformations: (A) –x − 1, −y, −z + 2; (B) –x − 1, −y + 1, −z + 2; (C) x − 1, y, z. (**b**) A drawing showing the **bey** topology. (**c**) A drawing showing the **sql** topology.

**Figure 5 molecules-28-02226-f005:**
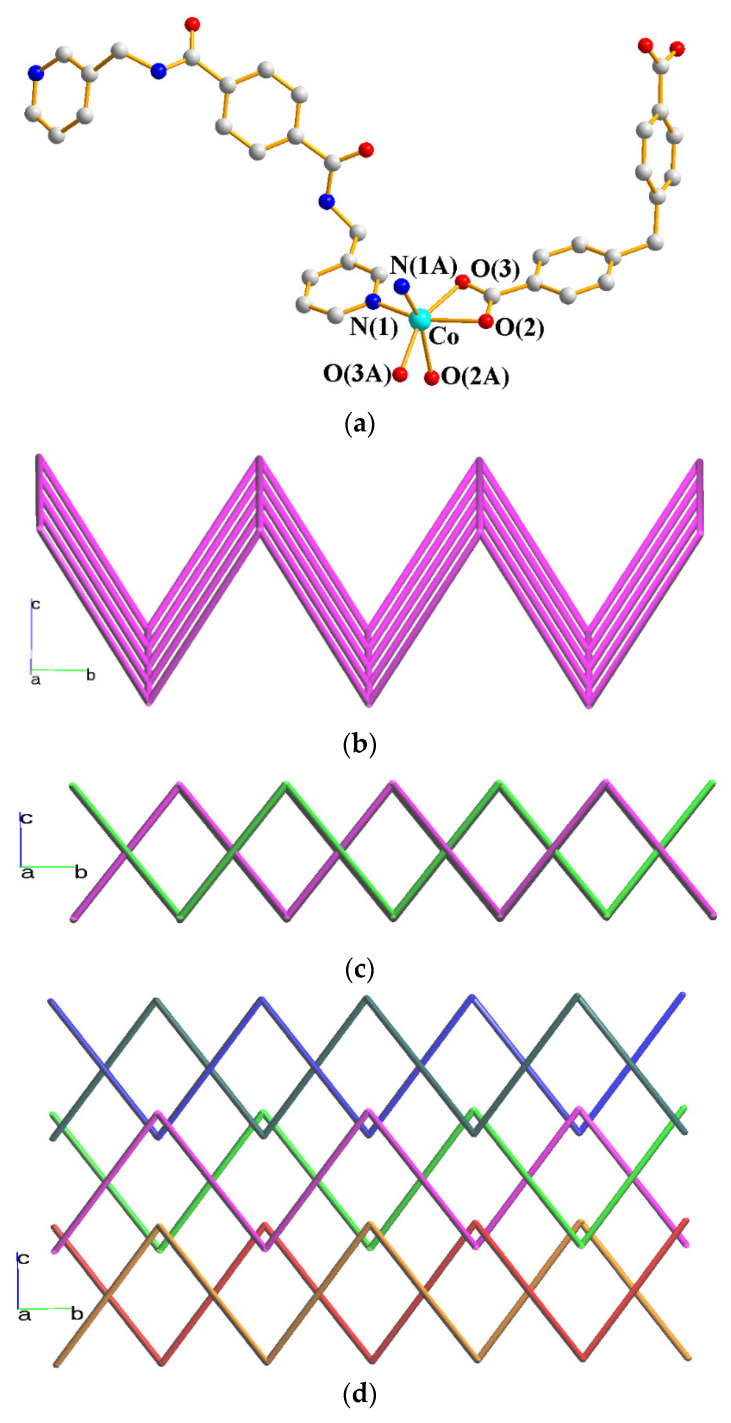
(**a**) Coordination environment around the Co(II) cation in **4**. Symmetry transformations used to generate equivalent atoms: (A) –x + 1, −y + 3/2, z. (**b**) A drawing showing the pleated 2D layer. (**c**) A drawing showing the 2-fold interpenetrated layers. (**d**) A drawing showing the polycatenation of 2-fold interpenetrated 2D nets.

**Figure 6 molecules-28-02226-f006:**
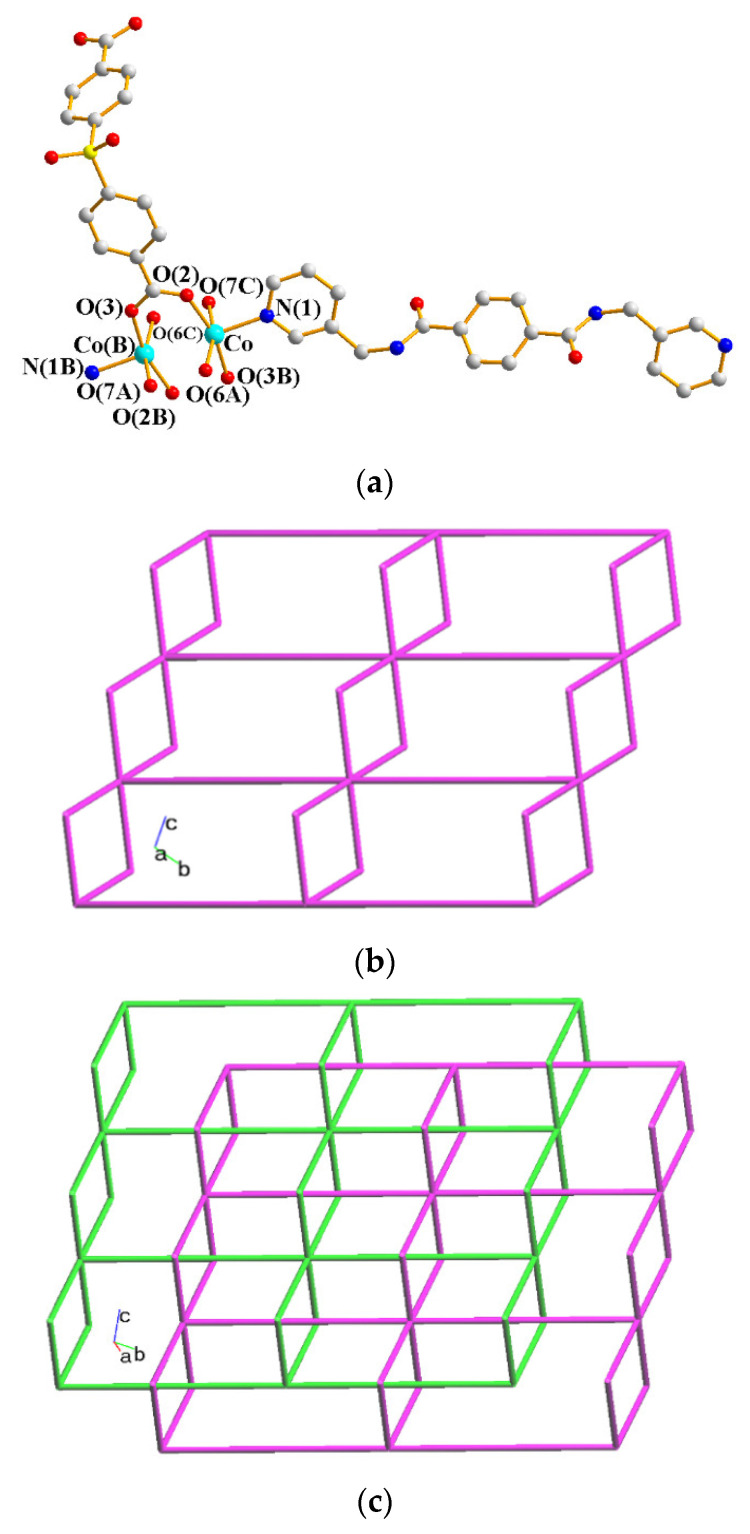
(**a**) The coordination environment of the Co(II) ion in **5**. Symmetry transformations used to generate equivalent atoms: (A) x, y, z − 1; (B) –x + 2, −y + 2, −z + 2; (C) –x + 2, −y + 2, −z + 3. (**b**) A schematic drawing showing the 2D layer with the 2,6L1 topology. (**c**) A schematic drawing showing the 2-fold interpenetrated 2D net.

**Figure 7 molecules-28-02226-f007:**
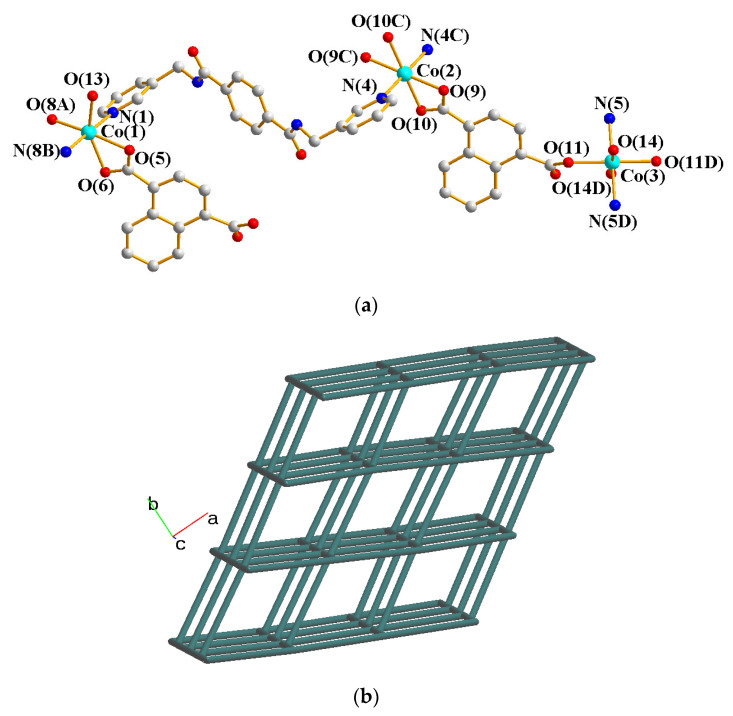
(**a**) The coordination environment of the Co(II) ion in **6**. Symmetry transformations used to generate equivalent atoms: (A) x, −y + 3/2, z + 1/2; (B) –x + 1, y − 1/2, −z − 1/2; (C) –x + 2, −y + 2, −z − 1; (D) –x + 2, −y + 2, −z − 2. (**b**) A drawing showing the **cds** topology.

**Figure 8 molecules-28-02226-f008:**
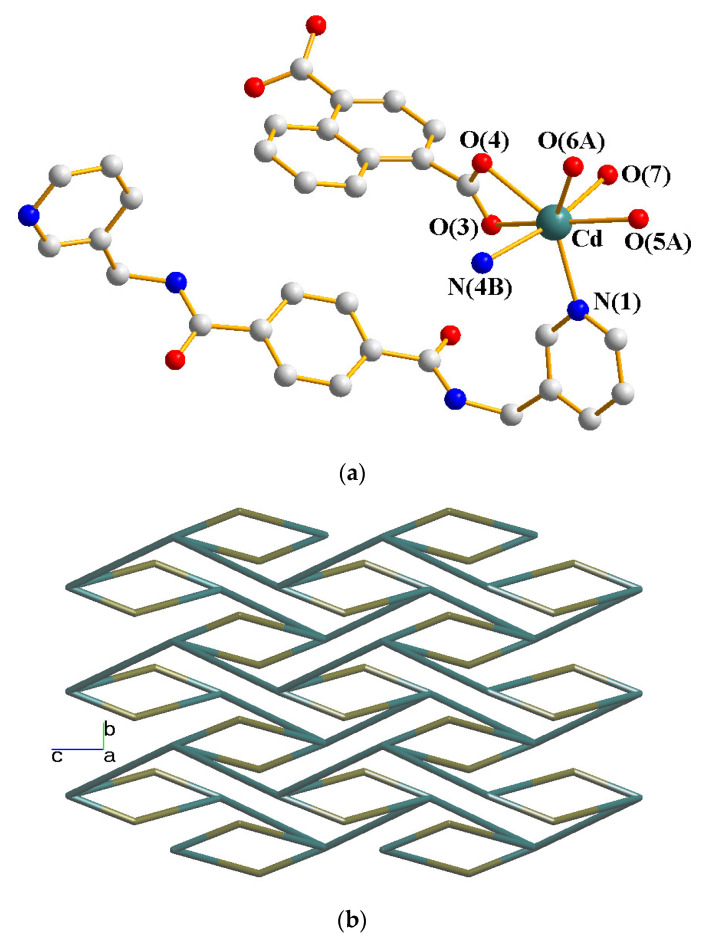
(**a**) The coordination environment of the Co(II) ion in **7**. Symmetry transformations used to generate equivalent atoms: (A) x, −y + 3/2, z + 1/2; (B) –x + 1, −y + 1, −z + 1. (**b**) A drawing showing the 2,4L1 topology.

**Figure 9 molecules-28-02226-f009:**
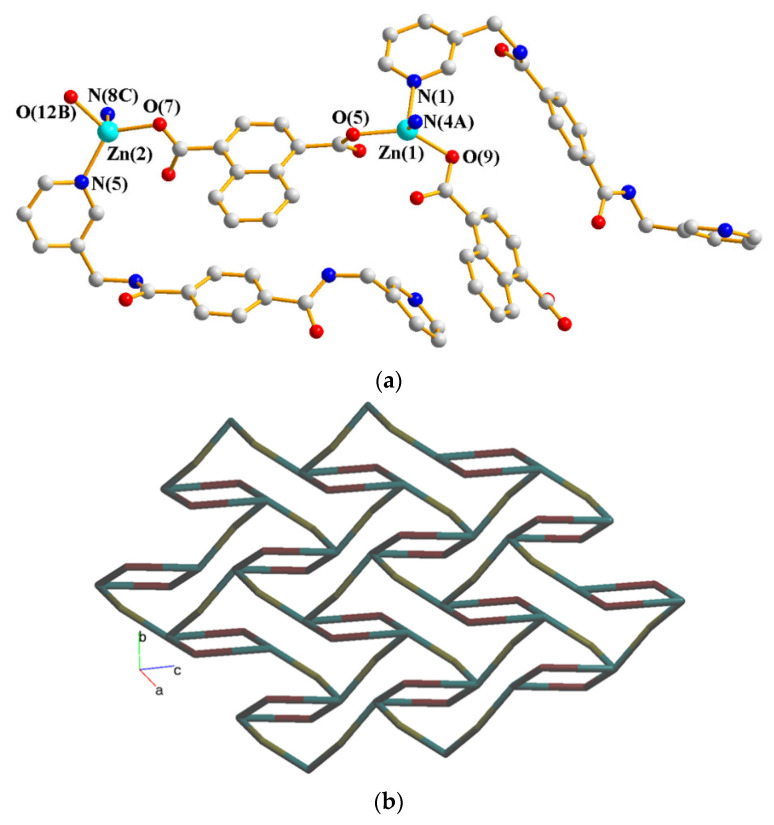
(**a**) The coordination environment of the Co(II) ion in **8**. Symmetry transformations used to generate equivalent atoms: (A) −x, −y + 2, −z + 2; (B) x, y, z – 1; (C) -x, −y + 1, −z + 1. (**b**) A drawing showing the (10^2^⋅12)(10)_2_(4⋅10⋅12^4^)(4) topology.

**Figure 10 molecules-28-02226-f010:**
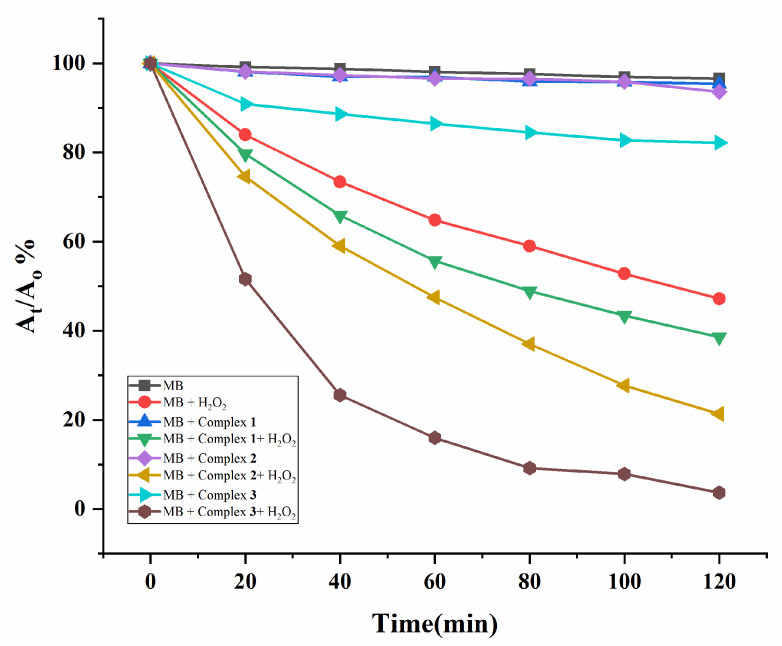
Photodegradation rates of MB solution under UV irradiation.

**Table 1 molecules-28-02226-t001:** Ligand conformations and bonding modes of **1**–**8**.

	Conformation	Coordination Mode
**1**	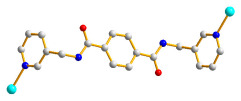 *t* *rans anti–anti* 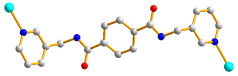 *trans anti–anti*	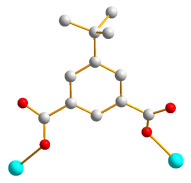 *μ_2_*-*κ*O:*κ*O′
**2**	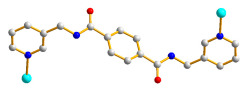 *trans anti–anti* 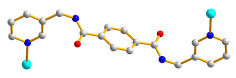 *trans syn*–*syn*	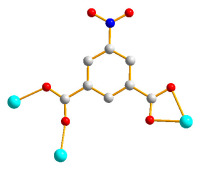 *μ_3_-κ^2^O,O′:κO′′:κO′′′*
**3**	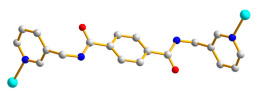 *trans anti*–*anti*	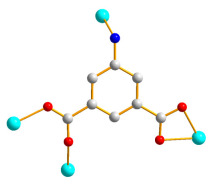 *μ_4_*-*κ^2^O,O′:κO′′:κO′′′*:*κ*N
**4**	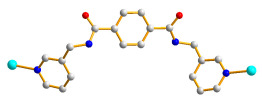 *cis syn–syn*	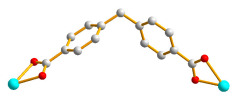 *μ_2_*-*κ^2^O,O′*: *κ^2^O′′,O′′′*
**5**	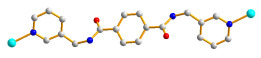 *trans anti*–*anti*	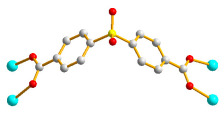 *μ_4_*-*κ*O:*κ*O′:*κ*O′′:*κ*O′′′
**6**	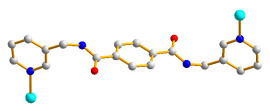 *trans syn*–*syn* 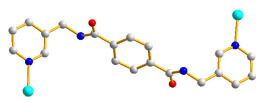 *trans anti*–*anti*	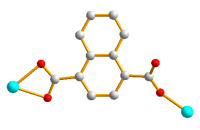 *μ_2_*-*κ^2^O,O′*:*κ*O′′
**7**	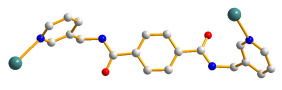 *trans syn*–*syn*	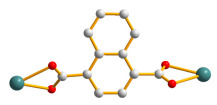 *μ_2_*-*κ^2^O,O′*: *κ^2^O′′,O′′′*
**8**	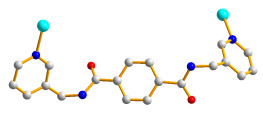 *trans syn*–*anti* 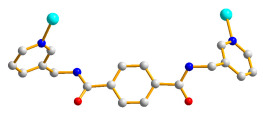 *cis anti–anti*	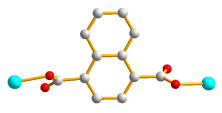 *μ_2_*-κO:κO′

**Table 2 molecules-28-02226-t002:** Crystallographic data for **1**–**8**.

Compound	1	2	3
Formula	C_32_H_34_N_4_O_8_Co	C_28_H_25_N_5_O_10_Co	C_19_H_18_CoN_3_O_6_
Formula weight	661.56	650.46	443.29
Crystal system	Triclinic	Triclinic	Triclinic
Space group	*P*ī	*P*ī	*P*ī
a, Å	10.3115(3)	10.0958(2)	8.3283(3)
b, Å	11.9684(4)	11.3920(3)	10.0484(3)
c, Å	13.2748(4)	12.6159(3)	11.3130(4)
α, °	74.6853(9)	73.5612(11)	81.041(2)
*β*, °	73.1745(9)	84.2220(10)	87.225(25)
γ, °	73.1501(9)	87.1266(11)	82.0898(19)
V, Å^3^	1471.49(8)	1384.23(6)	925.94(5)
Z	2	2	2
D_calc_, Mg/m^3^	1.493	1.561	1.590
F (000)	690	670	456
µ (Mo K_α_), mm^−1^	0.644	0.689	0.971
Range (2θ) for data collection, deg	3.270 ≤ 2θ ≤ 56.854	3.380 ≤ 2θ ≤ 56.662	3.646 ≤ 2θ ≤ 56.688
Independent reflections	7357 [R(int) = 0.0247]	6880 [R(int) = 0.0226]	4607 [R(int) = 0.0329]
Data/restraints/parameters	7357/0/406	6880/0/397	4607/0/262
Quality-of-fit indicator ^c^	1.035	1.065	1.005
Final R indices [I > 2σ(I)] ^a,b^	R1 = 0.0303 wR2 = 0.0732	R1 = 0.0536 wR2 = 0.1556	R1 = 0.0378 wR2 = 0.0824
R indices (all data)	R1 = 0.0369, wR2 = 0.0762	R1 = 0.0656, wR2 = 0.1659	R1 = 0.0548, wR2 = 0.0895
**Compound**	**4**	**5**	**6**
Formula	C_35_H_32_CoN_4_O_8_	C_24_H_19_CoN_2_O_8_S	C_64_H_62_Co_2_N_8_O_19_
Formula weight	695.57	554.40	1365.07
Crystal system	Orthorhombic	Triclinic	Monoclinic
Space group	*Ibca*	*P*ī	*P*2_1_/*c*
a, Å	13.6153(3)	9.1565(4)	16.5329(3)
b, Å	19.3800(5)	10.9657(5)	17.1867(3)
c, Å	25.1236(6)	13.2184(6)	21.3467(4)
α, °	90	97.8126(15)	90
*β*, °	90	109.4504(17)	103.3525(9)
γ, °	90	104.9062(15).	90
V, Å^3^	6629.2(3)	1173.06(9)	5901.61(19)
Z	8	2	4
D_calc_, Mg/m^3^	1.394	1.570	1.536
F (000)	2888	568	2832
µ (Mo K_α_), mm^−1^	0.576	0.874	0.649
Range (2θ) for data collection, deg	3.242 ≤ 2θ ≤ 52.000	3.962 ≤ 2θ ≤ 56.714	3.076 ≤ 2θ ≤ 56.588
Independent reflections	3267 [R(int) = 0.0767]	5836 [R(int) = 0.0234]	14613 [R(int) = 0.0466]
Data/restraints/parameters	3267/1/223	5836/0/334	14613/0/841
Quality-of-fit indicator ^c^	1.014	1.067	1.060
Final R indices [I > 2σ(I)] ^a,b^	R1 = 0.0492, wR2 = 0.1109	R1 = 0.0375, wR2 = 0.1079	R1 = 0.0526, wR2 = 0.1264
R indices (all data)	R1 = 0.1309, wR2 = 0.1409	R1 = 0.0493, wR2 = 0.1155	R1 = 0.0892, wR2 = 0.1448
**Compound**	**7**	**8**	
Formula	C_32_H_30_CdN_4_O_9_	C_64_ H_52_Zn_2_N_8_O_14_	
Formula weight	727.00	1287.87	
Crystal system	Monoclinic	Triclinic	
Space group	*P*2_1_/*c*	*P*ī	
a, Å	16.9732(3)	a = 12.8606(12)	
b, Å	9.4163(2)	b = 13.1507(14)	
c, Å	20.7184(4)	c = 18.7726(18)	
α, °	90	90.177(6)	
*β*, °	114.0812(10)	101.414(6)	
γ, °	90	113.475(5)	
V, Å^3^	3023.12(10)	2842.7(5)	
Z	4	2	
D_calc_, Mg/m^3^	1.597	1.505	
F(000)	1480	1328	
µ(Mo K_α_), mm^−1^	0.786	0.922	
Range (2θ) for data collection, deg	2.628 ≤ 2θ ≤ 56.620	2.222 ≤ 2θ ≤ 56.840	
Independent reflections	7521 [R(int) = 0.0571]	14194 [R(int) = 0.0806]	
Data/restraints/parameters	7521/0/419	14194/0/793	
Quality-of-fit indicator ^c^	1.009	1.004	
Final R indices [I > 2σ(I)] ^a,b^	R1 = 0.0373, wR2 = 0.0670	R1 = 0.0567, wR2 = 0.1012	
R indices (all data)	R1 = 0.0672, wR2 = 0.0763	R1 = 0.1485, wR2 = 0.1265	

^a^ R_1_ = F_o_ − F_c_/F_o_. ^b^ wR_2_ = [w(F_o_^2^ − F_c_^2^)^2^/w(F_o_^2^)^2^]^1/2^. w = 1/[σ^2^(F_o_^2^) + (ap)^2^ + (bp)], *p* = [max(F_o_^2^ or 0) + 2(F_c_^2^)]/3. a = 0.0329, b = 0.9214, **1**; a = 0.0873, b = 1.5032, **2**; a = 0.0413, b = 0.5057, **3**; a = 0.0644, b = 0.1557, **4**; a = 0.0665, b = 0.5236, **5**; a = 0.0590, b = 6.0526, **6**; a = 0.0259, b = 1.0874, **7**; a = 0.0475, b = 0.1495, **8**. ^c^ quality-of-fit = [Σw(|F_o_^2^| − |F_c_^2^|)^2^]/(N_observed_ − N_parameters_)^1/2^.

## Data Availability

Data are contained within the article or Appendix A.

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
