# Peer review of "Metal and Ligand Effect on the Structural Diversity of Divalent Coordination Polymers with Mixed Ligands: Evaluation for Photodegradation"

_molecules, 2023, doi:10.3390/molecules28052226_

Round 1

Reviewer 1 Report

The manuscript reported the syntheses, structures and photodegradation properties of a series of mixed ligand coordination complexes. The manuscript is interesting, thus I recommended the publication of the manuscript after some revisions.

1.      The figures of the manuscript are only arranged from top to bottom. For example, the figures are arranged in a sequence a, b, c, d, and e, you can rearrange a and b parallel to each other and then c and d parallel to each other and so on.

2.      Please make a comparison among the 8 compounds, and deleted all the repetitive descriptions.

3.      Please make a more detailed discussion about the photodegradation. Can you give the mechanism of the catalysis?

Reviewer 2 Report

Coordination polymers and metal-organic frameworks are one of the hot spots in coordination chemistry. Such compounds are considered as sensors, separators of substances with similar physical properties, luminescent, catalytic materials, etc.

Current work presents eight different coordination polymers of divalent metals with N,N'-bis(pyridin-3-ylmethyl)terephthalamide and some dicarboxylic acids. The compounds were characterized by elemental analysis, IR spectroscopy, and powder XRD data The structure of all compounds was established by single-crystal XRD. Three complexes were tested in a model reaction of methylene blue (MB) photodegradation.

In general, the manuscript presents new results and may be of interest to the audience of Molecules, especially within the framework of the Special Issue. However, a number of significant improvements are required.

1) The most significant drawback is the fragmentation of this work from the entire large field of research in the field of coordination polymers

a) It starts with the Introduction, when existing works with N,N'-bis(pyridin-3-ylmethyl)terephthalamide are not discussed.

b) Further, each structure of the complex is considered fragmentarily. There are no attempts to compare with related compounds and identify some common patterns. For example, how does the metal center and its environment affect the conformation N,N'-bis(pyridin-3-ylmethyl)terephthalamide, and conformation to structural type.

c) Finally, there are no sufficient comparisons for BET surface areas and observed catalytic effects.

Thus, it is impossible to determine what new and valuable this work has brought to the area under consideration, except for a simple increase in the number of synthesized compounds.

It is also possible to evaluate further directions of work or areas of application in which some of the obtained complexes will be promising due to their special structure.

2) The conclusion that complex 3 exhibits the most efficiency towards MB degradation does not correspond to the results of the work. In fact, there is a change in the powder diffraction pattern 3 after photodegradation experiments. Thus, it is quite probable that complex 3 decomposes, and the significantly increased activity was due to the decomposition products. This point should be made clear.

a) It should be assumed what could happen to the complex during the experiments. For example, check the elemental composition of a well-purified sample after the experiment, compare the IR spectrum. Check the degree of oxidation of cobalt. Growing sample crystals after the experiment is also encouraged.

b) It should be clarified whether such degradation occurs simply by irradiation. Perhaps this is a light-induced phase transition?

c) It should be clarified whether the diffraction patterns change in the same way if photoactivation is carried out in the presence or absence of hydrogen peroxide. Perhaps there is a partial (complete?) oxidation of cobalt (II)?

3) A brief discussion of the chosen synthetic strategy and IR spectra (most characteristic peaks) should be added. What can be the reason for the formation of complexes with different metal:ligand ratios (compounds 2 vs. 3)?

4) The elemental analysis data for complex 5 should be considered unsatisfactory (Calculated: N, 5.05%. Found: N, 6.54%). Please explain the reason. Perhaps this is due to the variability of the composition due to the lability of water molecules?

5) The addition of data on the thermal stability and solubility of the studied complexes is welcome.

6) Modifications made to the N,N'-bis(pyridin-3-ylmethyl)terephthalamide (L) preparation procedure should be clearly labeled, as should their effect (increased yield, shortened synthesis time, convenience?). Primary characterization data for this compound should also be added.

7) The description of the synthesis of complex 7 (p. 3.3.7) should be corrected. In fact, complex 7 cannot be "prepared as described for 7".

8) Technical points:

а) Please indicate elemental analysis accuracy (analyzer performance). Usually, the accuracy here does not exceed 0.5%. Therefore, it is more correct to represent the data in the form "58.8" than "58.78". Correct further.

b) Designations should be unified. For example, acetate is referred to in the text as "OAc" or "CH3COO". Choose one designation. The same applies to CH3OH/MeOH, 1,4-NDC2-/1,4-NDC, etc.

с) The procedure for conducting catalytic tests should be transferred to the Experimental part with an indication of the equipment used. Also give the absorption spectra of the corresponding complexes. Indicate whether the complexes dissolve during the experiments.

Round 2

Reviewer 1 Report

Dear editor,

        The manuscript can be accepted now.

Author Response

We sincerely appreciate the reviewer’s valuable comments.

Reviewer 2 Report

The authors made some useful changes to the manuscript. However, a number of points still require improvement. Moreover, there are moments that do not allow this article for publication.

1) The most principal moment is the lack of persuasiveness in preserving the composition of crystals 3 after catalytic experiments. In fact, the powder XRD does not give any information about the composition of the connection, and also does not prove that this is a unit phase, not a mixture of phases. The data of elemental analysis and IR spectra of samples 3 after the testing should be clearly shown at least. An additional convincing characterization is extremely desirable.

I mean that the change in the powder XRD pattern of 3 may not be caused by a structural transformation of complex, as the authors now believe, but by the decomposition of the complex or a (partial) oxidation of cobalt to Co3+.

Another reason may be a change in solvate composition, which the authors believe for 5 (note that also for a cobalt complex).

Additional data and their discussion should be submitted.

2) Data on a change in the composition and diffractograms of the complex 3 should be added if the photocatalytic tests (a) will not be involved in hydrogen peroxide (= only irradiation) (b) without involvement of irradiation (= only in the presence of oxidizer). This can help shed light on the cause of changes in crystals 3. In any case, the change in the composition /structure of the complex 3 after testing should be noted in the abstract /conclusions.

3) An explanation of inconsistency in the elemental composition of complex 5 is strange. In fact, analysis for all elements (C, H and N) is carried out from one sample, and not of different samples stored of different times. Therefore, the discrepancy between the results between (C, H) and N rather indicates the inconsistencies of the sample. In theory, this should be reflected in the powder XRD pattern. In any case, the explanation of the authors about the lability of the system indicating this example should be added to section 2.1.

4) A comparison of catalytic capabilities with literature should be more clearly represented in the form of a table. This table should also contain experimental data (concentration of MB, H2O2, etc.), otherwise the correctness of the comparison is unclear. This important information still has not been presented. Discussion and possible explanations should be presented in the text.

5) I insist that literary data on the MOFs with L-ligand as well as the functional properties investigated for them, should be presented in the Introduction. In fact, the Introduction should clearly outline the scientific landscape of the study, but now it is absent. Now it seems that the authors were the first to use this ligand. This is wrong. Thus, correct the introduction and more clearly formulate motivation to work.
